# It takes two for security: A Verifiable CO-Aggregation Protocol for Heterogeneous Federated Distillation

## Abstract

Federated distillation (FD) enables efficient collaboration among heterogeneous models, yet its rising application in privacy-sensitive fields raises security concerns. Advanced countermeasures have introduced secure aggregation protocols, which are broadly classified as centralized (server-dominated) or fully distributed (server-free). The former risks adversarial server interference, while the latter avoids central threats but suffers from poor energy efficiency due to coordination overhead. Neither approach offers an ideal solution. This paper introduces SVAFD, a secure co-aggregation protocol that ensures privacy protection and verifiability by redefining the responsibility boundaries between server and client in line with their respective requirements and resource capabilities. Specifically, SVAFD allows clients to perform lightweight encoding of logits and achieve locally partial aggregation, while the server converges all partial results for global decoding and generates aggregation proofs. By shifting from unilateral dominance to multilateral client/server co-aggregation, SVAFD guarantees that no vanilla clients' privacy can be inferred while providing with an unforgeable aggregation proof for process checking, even in the presence of server collusion with a subset of clients. Moreover, SVAFD is resilient to stragglers and reduces the complexity for each aggregation from three rounds (Armadillo, NeurIPS'24) or more in recent protocols to only two rounds, chieving a 1.32× reduction in computation consumption.

## 1 INTRODUCTION

Federated learning (FL) Duan et al. (2023); Yang et al. (2019); Yu & Li (2021); Zhao et al. (2022); McMahan et al. (2017) has gained widespread popularity in privacy-sensitive domains such as medicine Nasajpour et al. (2025) and remote sensing (Moreno-Alvarez et al., 2024), facilitating collaborative training across distributed datasets without exposing raw data (Letaief et al., 2022; Wu et al., 2024a). However, its scalability is hindered by the restrictive requirement of model homogeneity and the considerable communication overhead (Wu et al., 2023a; Shao et al., 2023). In this regard, Federated Distillation (FD) employs the transmission of model outputs, known as logits, in place of complete mdoel parametes (Hinton et al., 2015). This refinement not only reduces communication overhead, as the transmitted information scales with small output dimensionality rather than model size, but also allows clients to train heterogeneous and personalized models, thereby improving flexibility and efficiency (Wu et al., 2023b; Jin et al., 2023).

Although FD circumvent the direct disclosing of raw data or model updates, machine learning community reveals that the uploaded logits retain abundant information about the client's training data, to the extent that pixel-level sample reconstruction Takahashi et al. (2023) and analysis of feature distribution Shao et al. (2023) become possible. This exposure highlights the vulnerability of FD system to inference attacks (Li et al., 2024a; Tang et al., 2024).

**Challenges:** Conventional secure aggregation countermeasures, that ensure both privacy protection and process verifiability, are broadly divided into centralized protocols (server-dominated) Lyck-lama et al. (2023); Bonawitz et al. (2017); Rathee et al. (2023); Yu et al. (2024) and fully distributed (server-free) protocols (Wu et al., 2023c; Fan & Maoting, 2024). Centralized protocols designate

the server as the aggregation subject for key stages including logits aggregation and verification generation, while ensuring clients' privacy through uploading encrypted or perturbed logits (Bonawitz et al., 2017; Rathee et al., 2023; Li et al., 2024b). However, an adversarial server can deviate from the prescribed aggregation protocol to decrypt sensitive information, or collude with a subset of clients to generate deceptive aggregation proofs (Wu et al., 2023a; Yu et al., 2024; Jiang et al., 2021), thereby severely undermining the security guarantees that claimed.

In contrast, an alternative line of aggregation protocols assigns each client the autonomy to manage the aggregation process independently (Fan & Maoting, 2024). This fully decentralized method eliminates concerns of central trust, yet it incurs significant costs in coordination or global synchronization, which become even more pronounced when paired with mandatory verification mechanisms (Wu et al., 2023a; Ghavamipour et al., 2024). Moreover, such protocols tend to be highly sensitive to client dropouts or mismatches. Consequently, neither the server-dominated centralized aggregation nor the fully distributed client-driven aggregation provides an ideal solution (Kabir et al., 2024). We designate this challenge as the Aggregation Subject Dilemma.

**Motivation:** To address the above challenge, our intuitive idea is to redefine the responsibility boundaries between the server and the clients to realize complementary co-aggregation based on their respective security requirements and resource capacities. Specifically, the aggregation stages most relevant to privacy exposure should remain under client-side control, while the computationally intensive and coordination-oriented tasks are concentrated on the server side. The principal benefit of this reconstruction lies in shifting the aggregation process from the unilateral aggregation to Clients/Server multilateral co-aggregation, which breaks the dichotomy between existing centralized and fully distributed aggregation protocols, and establishes a balanced equilibrium between privacy protection and training efficiency.

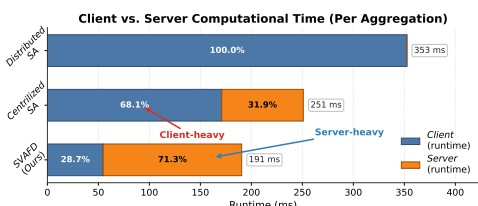

Figure 1: Computational runtime for different protocols. Distributed protocol Ghavamipour et al. (2024) suffers from low efficiency, while centralized method Ma et al. (2024) is vulnerable to central threats and heavy-computation for clients.

**Proposed defense:** In this paper, we propose a **S**ecure and **V**erifiabl co-**A**ggregation protocol for **FD** (SVAFD) to enable both privacy protection and verifiability. Specifically, clients share their encoded logits with each other via LCC Yu et al. (2019) and locally aggregate the collected logits shares, which are then sent to the server along with the corresponding logits signatures. The server decodes all the partially aggregated results to derive the global logits and aggregates all the signatures to provide each client with the verifiable proof for the correctness checking. SVAFD combines the advantages of both decentralized and centralized aggregation protocols. On the one hand, it preserves the server's role as the global coordinator and executor of computationally intensive tasks, such as signature aggregation and global decoding, thereby improving training efficiency; on the other hand, it draws inspiration from fully decentralized protocols by enhancing client autonomy, shifting the privacy-sensitive aggregation operation from server-side aggregation to client-side partial aggregation, thus minimizing the risk of privacy leakage. The overall overflow is illustrated in Figure 2. Our contributions can be summarized as:

- We study the secure aggregation in heterogeneous FD and identify the Aggregation Subject Dilemma that existing protocols over-rely on a single aggregation party (server or client), preventing a balanced trade-off between privacy protection and training efficiency.
- We introduce SVAFD, a secure and verifiable co-aggregation protocl that provides a principled balance between the existing centralized and fully distributed aggregation protocols. It redefines the server/clients' responsibility boundaries, where privacy-critical processing (logits aggregation) remains client-side, while the coordination server handles proof generation and computation-intensive tasks. SVAFD guarantees the preservation of vanilla clients' privacy, while remaining resilient to client dropouts.
- Across extensive experiments, SVAFD realizes a constant two-round aggregation protocol and requires 0.76× (Figure 1)the total computation of Armadillo's three-round scheme. Moreover, it achieves a 2.37× reduction in the proportion of client-side computation relative to the global time, indicating greater offloading of computation to the resource-rich server.

## 2 PRELIMINARIES

### 2.1 PROBLEM SETTING OF FD AGGREGATION

We consider federated distillation (FD) for a $D$-class task over $\mathcal{C}$ clients (Wu et al., 2023b). In each round $r$, client $c$ evaluates its local model on a subset of samples from the public/proxy dataset (Wu et al., 2024b), yielding the logits $\boldsymbol{\vartheta}_c \in \mathbb{R}^{O \times D}$, where $O$ denotes the number of distilled data[1].

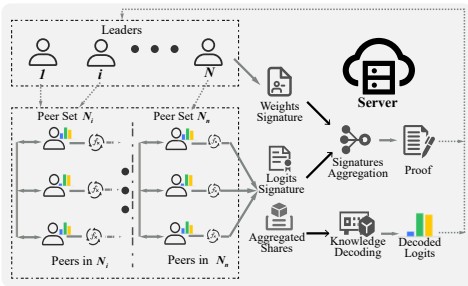

Figure 2: Illustration of SVAFD.

Unlike standard federated learning (FL) (Jeong et al., 2018a), where aggregation produces a single global object for all participants, heterogeneous FD must accommodate clients with differing model architectures (Shao et al., 2023). This motivates the use of client-specific aggregation neighborhoods[2] to aggregate customized teacher logits. Specifically, for each client $c$, the server selects a peer set $\mathcal{N}_c \subseteq \mathcal{C} \setminus \{c\}$ (not necessarily identical across clients), and computes: $Y_c = \hbar(\{\boldsymbol{\vartheta}_i : i \in \mathcal{N}_c\})$, where $\hbar(\cdot)$ denotes the aggregation operator. Throughout, for clarity of exposition, we fix a *leader* client $c$ with *peer* set $\mathcal{N}_c$ and use this pair as a running example, while taking the size in this instance to be $N \triangleq |\mathcal{N}_c|$. All definitions and results extend to any client in $\mathcal{C}$ by symmetry.

### 2.2 THREAT MODEL

We adopt the most commonly used FD setting with a single (logical) server (Shao et al., 2023). The adversary is probabilistic polynomial-time (PPT) and may adapt across rounds (Ma et al., 2024).

- **Attacker's goals.** (i) Deceive the benign clients into accepting tampered or poisoned logits (e.g., via forged proofs), thereby misleading local training; and (ii) infer private information about vanilla clients' logits beyond what is revealed by the intended aggregate.

- **Attacker's capabilities.** The adversary may corrupt an adaptively chosen subset $\mathcal{S} \subseteq \mathcal{N}_c$ of clients with $|\mathcal{S}| \leq \eta N$, where $\eta \in [0, 1)$ bounds the per-round corruption rate within $\mathcal{N}_c$. The server may also be corrupted and may collude with clients. Up to a $\delta$ fraction of clients may drop out per round, i.e., $|\mathcal{D}| \leq \delta N$ with $\delta \in [0, 1)$. The adversary can adversarially schedule these dropouts subject to the per-round budget.

- **Attacker's view.** The union of (i) internal states of corrupted peers $i \in \mathcal{S}$; (ii) internal states of corrupted server; and (iii) all messages sent/received by these parties. If the server is corrupted, the adversary can forge or fork the downlink destined for client $c$ and observe server-side auxiliary signals. Views accumulate over rounds and may inform future adaptive choices.

### 2.3 COMMUNICATION MODEL

Clients in our system are heterogeneous and may become temporarily unresponsive due to hardware or network fluctuations (Ma et al., 2024). An implicit distribution for clients' response times is assumed. All communications between a client and the server take place over authenticated and confidential channels. Messages exchanged among clients are forwarded via the server, as in prior works (Bell et al., 2020; Ma et al., 2024). We assume such secure client-to-client communication throughout the protocol. Details regarding the definition of the communication round are provided in Appendix B.

---

[1]Depending on the granularity of logits exchanged, existing FD approaches can be categorized as class-grained with $\boldsymbol{\vartheta}_c \in \mathbb{R}^{D \times D}$ and sample-grained with $\boldsymbol{\vartheta}_c \in \mathbb{R}^{O \times D}$, see Appendix A for details

[2]The specific client selection strategy is orthogonal to our secure aggregation protocol. In this work, we adopt random sampling to construct peer sets.

## 2.4 Security Guarantees

Owing to system and network limitations, our aggregation protocol is designed to deliver the following guarantees. Formal definitions are deferred in Theorem 3.1.

- **Privacy Protection.** Even in the presence of a corrupt server colluding with up to $T = \eta N$ clients, the privacy of vanilla clients' logits is preserved, provided that Theorem 3.1 holds for every selected peer set.
- **Dropout Recovery.** For the instability of client devices, some clients may disconnect during aggregation (either passively or actively). Unlike prior approaches that simply discard the corresponding data, our protocol employs redundancy coding to rapidly reconstruct the missing logits when up to $D = \delta N$ users drop out, allowing the protocol to proceed without interruption.
- **Verifiability.** The protocol provides clients with verifiable assurance of correct logits aggregation. A corrupt server or subset of clients, even if colluding cannot produce a proof that passes verification and misleads honest participants, unless all $N - 1$ clients collude.

## 3 Technical Overview

This section overviews how SVAFD achieves secure and verifiable co-aggregation.

### 3.1 MM-LCC Design for Co-aggregation

We start with a base secure aggregation scheme with only dropout resilience and privacy security. At a high level, we refactor unilateral aggregation, whether server-dominated or fully decentralized, into a two-stage co-aggregation pipeline: (i) client-side partial aggregation performed directly on encoded logits, and (ii) server-side global decoding that reconstructs the teacher logits from the partially aggregated results. This division keeps privacy-critical computation (encode and aggregation) on client-side while offloading coordination and heavy decoding tasks to the server, yielding constant two-round communication. Before presenting the technical details of our construction, we formally state the security properties guaranteed by SVAFD.

**Theorem 3.1** $((D, T, \hbar, I) - achievable$ of SVAFD.)**. *SVAFD enables secure co-aggregation for its property of D-Resilient, T-Privacy, $\hbar$-Operational and I-Verifiable, as long as*

$$D + deg(\hbar)(K + T - 1) + 1 \leq N \tag{1}$$

Given $N$ Clients and the degree $\deg(\hbar)$ of the aggregation function $\hbar(\cdot)$, $D$-Resilient donotes that the server can decode the global logits $\widetilde{Y}$ even with up to $D$ clients dropping out. This is because SVAFD inherently adopts the redundant encoding scheme, which will be described in detail later. $T$-Privacy eliminates the leakage of clients' privacy, demonstrating that no meaningful information about local logits $\vartheta_i, i \in \mathcal{N}_i$ can be inferred by the server or at most $T$ colluding clients. The proof of $T$-Privacy will be provided in Appendix K. $\hbar$-operational highlights any multi-party aggregation functions with the polynomial degree of $\deg(\hbar)$ are supported. It indicates that SVAFD not only accommodates common linear operations in FD with $deg(\hbar) = 1$, but is also extendable to scenarios requiring intricate polynomial mappings for high-dimensional or scale feature representations, such as general tensor algebra, gradient computation Shao et al. (2022) and bilinear computation (Yu et al., 2019). Additionally, the $\hbar$ aggregation function here is interactive, operating on multiple clients' logits shares, which differs from the $f$ function in vanilla LCC that only computes on a single data source. Ultimately, $I$-Verifiable refers to the property whereby SVAFD provides each client $i$ with proofs to verify the correctness of the aggregation process. These proofs are unforgeable unless all the $N$ peer clients collude with the server. We prove the $I$-Verifiable secure property in §J.

**Two Rounds of secure co-aggregation.** We realize the two-stage pipeline for co-aggregation via Multi-to-Multi Lagrange Coded Computing (MM-LCC), an extension of vanilla LCC (details in Appendix C)that enables multiple data providers (DPs) and multiple service processor (SPs) to collaboratively encode, aggregate, and decode logits in heterogeneous federated distillation setting. Under MM-LCC, each client plays a dual role, it is a DP contributing its own logits and, simultaneously, an SP that performs partial aggregation for others.

We instantiate the co-aggregation pipeline by first specifying the *aggregation operator*

$$\hbar(\mathcal{V}) = \mathcal{W}f(\mathcal{V}) \tag{2}$$

where $\mathcal{V} = [\boldsymbol{\vartheta}_1, \ldots, \boldsymbol{\vartheta}_N]$ and $\mathcal{W} = [w_1, \cdots, w_N]$ denote the local logits and the corresponding aggregation weights, respectively. Each $w_i \in \mathbb{R}_{\geq 0}$ is a normalized weight vector with $\sum_i w_i = 1$, and can be customized by individual clients to amplify or attenuate specific logits for training.

**Round-1 (client $\longleftrightarrow$ client).  i) Data preprocess:** By default we adopt the sample-grained[3] setting where $\boldsymbol{\vartheta}_i \in \mathbb{R}^{O \times D}$. We split along the sample axis into $K$ blocks to acquire $\overline{\boldsymbol{\vartheta}}_i \in \mathbb{R}^{K \times \Omega \times D}$, with each block $\overline{\vartheta}_i^{(k)} \in \mathbb{R}^{\Omega \times D}$ and $\Omega = O/K$[4]. Then, client $i$ samples a blinding stack $\overline{H}_i \triangleq [\overline{h}_i^{(1)}, \overline{h}_i^{(2)}, \ldots, \overline{h}_i^{(T)}]$ and forms the pre-encoding tensor as:

$$\begin{aligned} B(\overline{\boldsymbol{\vartheta}}_i) &\triangleq [\overline{\vartheta}_i^{(1)}, \ldots, \overline{\vartheta}_i^{(K)}] || [\overline{h}_i^{(1)}, \ldots, \overline{h}_i^{(T)}] \\ &\triangleq [\overline{\vartheta}_i^{(1)}, \overline{\vartheta}_i^{(2)}, \ldots, \overline{\vartheta}_i^{(K)}, \overline{h}_i^{(1)}, \ldots, \overline{h}_i^{(T)}] \end{aligned} \tag{3}$$

where $\overline{h}_i^{(t)} \in \mathbb{C}^{\Omega \times D}$ is independently sampled from a zero-mean circularly symmetric complex gaussian distribution, with truncated standard deviation $[-\theta \frac{\sigma_i}{\sqrt{T}}, \theta \frac{\sigma_i}{\sqrt{T}}]$. The parameter $\sigma_i$ denotes the matrix variance, and $\theta$ is the truncation coefficient.

By employed the Analog LCC (Soleymani et al., 2021) encoding with Lagrange polynomials, the *encoded share* from client $i$ to recipient $j$ is

$$\widetilde{\vartheta}_{i \to j} = \sum_{k=1}^{K} \overline{\vartheta}_i^{(k)} l_k(\alpha_j) + \sum_{t=K+1}^{K+T} \overline{h}_i^{(t-K)} l_t(\alpha_j) \tag{4}$$

where $l_z(\alpha) = \prod_{l \in [K+T] \setminus \{z\}} \frac{\alpha - \beta_l}{\beta_z - \beta_l}$ for all $z \in [K+T]$. Additionally, the parameters $\alpha$ and $\beta$ are picked to be equally spaced on the circle of radius centered around 0 in the complex plane by $\alpha_i = \gamma^{i-1}, i \in \mathcal{N}_c$ and $\beta_l = \omega^{l-1}, l \in [K+T]$, where $\gamma = e^{-\frac{2\pi\iota}{N}}$ and $\omega = e^{-\frac{2\pi\iota}{K+T}}$ are the $N$-th and $(K+T)$-th roots of unity, respectively, with $\iota^2 = -1$. These shares are then exchanged among clients in $\mathcal{N}_c$.

**ii) Partial aggregation:** Each recipient $j \in \mathcal{N}_c$ will receive the logits shares set as $\widetilde{\boldsymbol{\vartheta}}_{\to j} = [\widetilde{\vartheta}_{1 \to j}, \widetilde{\vartheta}_{2 \to j}, \cdots, \widetilde{\vartheta}_{n \to j}] \in \mathbb{C}^{N \times \Omega \times D}$, and applies the share-level aggregator $\hbar : \mathbb{C}^{N \times \Omega \times D} \to \mathbb{C}^{\Omega \times D}$ as:

$$\hbar(\widetilde{\boldsymbol{\vartheta}}_{\to j}) = \widetilde{W} f(\widetilde{\boldsymbol{\vartheta}}_{\to j}) \tag{5}$$

where $\widetilde{W} \in \mathbb{R}^{N \times (\Omega \times D)}$ is the blinded weights initialized by the leading client $c$ detailed in Appendix F.

**Round-2 (client $\longleftrightarrow$ server)  i) Global decoding:** The server decodes the global logits by collecting sufficient numbers of aggregation results from each clients. For clients in peer set $\mathcal{N}_c$, each of the the aggregated sharings $\hbar(\widetilde{\vartheta}_{\to i}), i \in \mathcal{N}_c$ corresponds to an evaluation of $\hbar(u(\alpha_i)), i \in \mathcal{N}_c$, and the $u(\cdot)$ represents as

$$u(\alpha_i) = \sum_{k=1}^{K} \overline{\mathcal{V}}^{(k)} l_k(\alpha_i) + \sum_{t=K+1}^{K+T} \overline{\mathcal{H}}^{(t-K)} l_t(\alpha_i) \tag{6}$$

where

$$\overline{\mathcal{V}}^{(k)} = [\overline{\vartheta}_1^{(k)}; \overline{\vartheta}_2^{(k)}; \cdots; \overline{\vartheta}_N^{(k)}] \in \mathbb{R}^{N \times \Omega \times D}$$

$$\overline{\mathcal{H}}^{(t-K)} = [\overline{h}_1^{(t-K)}; \overline{h}_2^{(t-K)}; \cdots; \overline{h}_z^{(t-K)}] \in \mathbb{C}^{N \times \Omega \times D}$$

The decoding process consists of two steps. Firstly, the server needs to recover the coefficients of the polynomial $\hbar(u(\cdot))$. Since the degree of this composite polynomial is $deg(\hbar)(K+T-1)$, the server

---

[3]For completeness, we also support a class-grained verison in Appendix E.

[4]We assume that $O$ is divisible by $K$.

requires to receive at least $deg(f(x))(K + T - 1) + 1$ aggregated shares (evaluation points) from each peer set $\mathcal{N}_c$ to interpolate and construct the polynomial, where $deg(\hbar) = deg(f(x))$. Secondly, the server evaluates the polynomial $\hbar(u(z))$ at point $z = \beta_k, k \in [K]$ to obtain $\overline{Y}_c^{(k)} = \hbar(\overline{\mathcal{V}}^{(k)})$. Then the global knowledge split $[\overline{Y}_c^{(1)}, \overline{Y}_c^{(2)}, \cdots, \overline{Y}_c^{(K)}]$ can be decoded once the server has received sufficient aggregated shares, and the $D$-Resilient property claimed in Theorem 3.1 is proved.

Finally, the server reassembles the decoded blocks into a single tensor by extracting the real part:

$$\widetilde{Y}_c = \begin{cases} \sum_{k=1}^K \overline{Y}_c^{(k)}, & \text{class-grained}: \mathbb{C}^{K \times \Omega \times D} \to \mathbb{R}^{D \times D}, \\ \overline{Y}_c^{(1)} \| \cdots \| \overline{Y}_c^{(K)}, & \text{sample-grained}: \mathbb{C}^{K \times \Omega \times D} \to \mathbb{R}^{O \times D}. \end{cases} \tag{7}$$

**Lemma 3.2** (Homomorphic Share Aggregation). *MM-LCC is utilized to enable co-aggregation for its property of Homomorphic Share Aggregation. Specifically, any operation performed by the client on the encoded partial shares, after decoding, must correspond to the direct operation on the original logits:*

$$\hbar(\mathcal{V}) = \text{Dec}(\hbar(\text{Enc}(\mathcal{V}))) \tag{8}$$

where the $Enc(\cdot)$ and $Dec(\cdot)$ represent the LCC encoding and decoing process, respectively. Lemma 3.2 provides a theoretical basis for end-side partial aggregation and server-side global decoding. It indicates that the global logits $\hat{Y} = \hbar(\mathcal{V})$ obtained through decoding is equivalent to the ground truth value $Y^* = \mathcal{W}f(\mathcal{V})$, illustrated in the green color of Fig. C(d). The detailed proof can be seen in Appendix I.

Figure C provides a clear overview of the entire dataflow, illustrating how local logits, and aggregation are composed into the global co-aggregation pipeline. The key challenge remaining is achieving verifiability for aggregation process checking, which we discuss next.

### 3.2 VERIFIABLE CO-AGGREGATION

**Tripartite Proofs.** We instantiate verification for SVAFD in a *tripartite* setting aligned with our system design:

- **Leader $c$.** Provides the aggregation weights $\mathcal{W}$ for peers in $\mathcal{N}_c$; expects the aggregated logits $Y = \hbar(\mathcal{V})$ and the proof for correctness checking.
- **Peers $i \in \mathcal{N}_c$.** Provide local logits $\vartheta_i$ and perform the share-level partial aggregation $\hbar(\cdot)$.
- **Server.** Provides globally decoded logits and the aggregated proof $\Pi_c$ to the leader.

We adopt the framework based on matrix digests (Liu et al., 2023) and bilinear pairings: peers and the server act as provers that the weighted aggregation and decoding are executed as prescribed, while the leader verifies the returned logits and proof $\Pi_c$.

**Public parameters and one-time blinding.** Let $(p, \mathbb{G}, \mathbb{G}_T, e)$ be bilinear groups of order $p$ with generator $g$. Let $\nu \in \{0, 1\}^{|D_{pub}| \cdot D}$ define the *matrix digest* (a linear map compressing a logit matrix to a scalar); we write $\text{md}(\cdot)$ for digesting by $\nu$.

**Round 1 (client $\longleftrightarrow$ client): Signatures for weights and logits** The leader $c$ and peers in $\mathcal{N}_c$ produce auxiliary signatures for later verification. For peer $i \in \mathcal{N}_c$, with splitted logits $\overline{\vartheta}_i = [\overline{\vartheta}_i^{(1)}, \ldots, \overline{\vartheta}_i^{(K)}]$, we get the blockwise digests $V_i^{(k)} := \text{md}(\overline{\vartheta}_i^{(k)})$ (implemented as the blockwise element-sum in our default), and compute

$$\pi_i^{(k)\prime} = g^{V_i^{(k)} + \Upsilon_i}, \qquad i \in \mathcal{N}_c, \ k \in [K], \tag{9}$$

where $\Upsilon_i$ is the follower's private key used to detect misbehavior during share aggregation. The leader signs the weights $\mathcal{W}_c = [w_1, \ldots, w_N]$ by

$$\pi_i'' = g^{w_i}, \qquad i \in \mathcal{N}_c. \tag{10}$$

Finally, the peers and leader upload the signatures $\pi_{\mathcal{V}}^c = \{\pi_1', \ldots, \pi_N'\}$ (with $\pi_i' = [\pi_i^{(1)\prime}, \ldots, \pi_i^{(K)\prime}]$) and $\pi_{\mathcal{W}}^c = \{\pi_1'', \ldots, \pi_N''\}$ to the server, alongside the partial aggregates produced in Round 1. Moreover, each peer $i \in \mathcal{N}_c$ transmits its private key $\Upsilon_i$ to the leader, enabling

the subsequent verification procedure. Due to the fact that most signature computation approaches are designed based on operations over large integers, we adopt an approximate mapping function to convert floating-point values $V_i^{(k)}$ and $w_i$ to suitable integer domains. The details are provided in Appendix G.

**Round 2 (client $\longleftrightarrow$ server): global decoding and signature aggregation.** The server globally decodes the received partial aggregates to obtain the masked aggregate $\widetilde{Y}_c$, shown in § 3.1. It then aggregates pairings using the uploaded signatures:

$$\Pi_c = \prod_{i \in \mathcal{N}_c} e\Big( \prod_{k=1}^{K} \pi_i^{(k)\prime}, \pi_i'' \Big) \in \mathbb{G}_T, \tag{11}$$

and returns the tuple $Res_c = (\widetilde{Y}_c, \Pi_c)$ to the leader $c$. Implementation notes of Bilinear Pairings appear in Appendix H. Next, the masking factor is removed element-wise:

$$\hat{Y}_c = \mathbf{R}_c^{-1} \odot \widetilde{Y}_c, \tag{12}$$

where $\mathbf{R}_c$ is the mask matrix and its construction and invertibility conditions are given in Appendix F. Finally, the leader verifies a *single pairing equation*:

$$e(g,g)^{\sum_{g=1}^{M} \sum_{l=1}^{D} y_{g,l} + K \sum_{i \in \mathcal{N}_c} w_i \Upsilon_i} \stackrel{?}{=} \Pi_c, \tag{13}$$

where $y_{g,l}$ is the $(g,l)$-th entry of $\hat{Y}_c$, and $M = D$ (class-grained) or $M = O$ (sample-grained). Because the digest is linear and signatures are bound to round tags, any deviation by the server or peers (tampering logits, shares, or weights) breaks equation 13. This keeps verifier cost low enough for resource-constrained clients while avoiding a trusted third party. The correctness and non-forgeability of the above process can be found in Appendix J, and the pseudo-code can be found in Appendix N.

## 4    IMPLEMENTATION AND EVALUATION

To evaluate the design and implementation of SVAFD, we ask the following essential questions:

- What is the breakdown of performance of individual components in our protocol?
- How does SVAFD compare to state-of-the-art aggregation solutions?

### 4.1    PERFORMANCE BREAKDOWN OF SVAFD

In this section, we evaluate the system-level overhead of SVAFD, including experiments on the client side (leader and peer) as well as on the server side across different stages. The specific stage division can be found in Appendix L. We present the results in Fig. 3. In the FD setting, the aggregation parameter size is determined by the number of distilled samples per round[5]. Fixing the batch size at $bs = 32$, we examine the latency under varying $K$ values of 20, 30, and 40.

**Runtime analysis.** As shown in Fig. 3(a), the peer-side initialization latency mainly depends on the number of clients, while auxiliary computation grows with $K$ and can be nearly ten times higher. This overhead stems from signing operations, yet even with $K = 40$ and $N = 200$ clients, latency remains within milliseconds. Fig. 3(b) illustrates that leader-side initialization overhead rises with both $N$ and $K$, whereas auxiliary computation is dominated by quantization and signature and scales mainly with $N$. The leader's total overhead grows nearly linearly with the number of clients.

Fig. 3(c) show that encoding/decoding overhead increases linearly with $K$ and $N$, consistent with the lightweight linear nature of LCC Yu et al. (2019). As seen in Fig. 3(d), the server proof generation is the most time time-consuming. This demonstrates that relying solely on resource-constrained clients in the decentralized aggregation protocols to perform the signature aggregation required for

---

[5]The number of distilled samples is defined as $\Omega = K \times bs$. Consequently, the aggregation parameter size can be computed as $\Omega \times D$, where $D$ denotes the total number of classes.

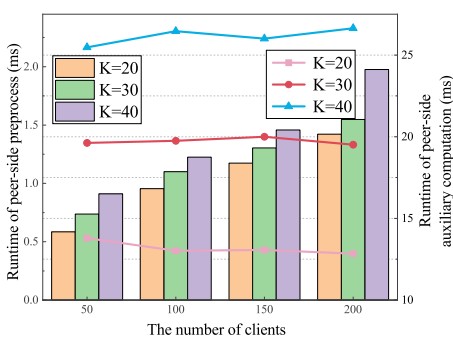

(a) Runtime of preprocess and auxiliary for peers.

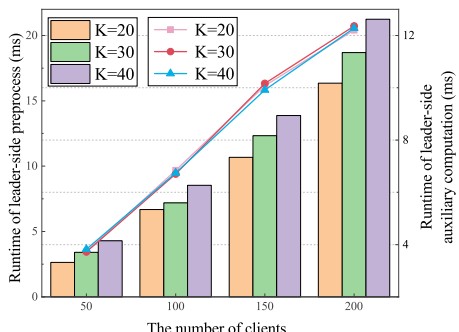

(b) Runtime of preprocess and auxiliary for leaders.

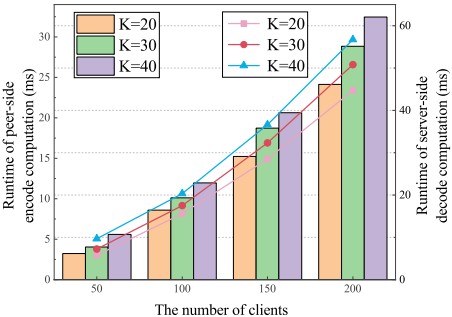

(c) Runtime of encode/decode for peers and server.

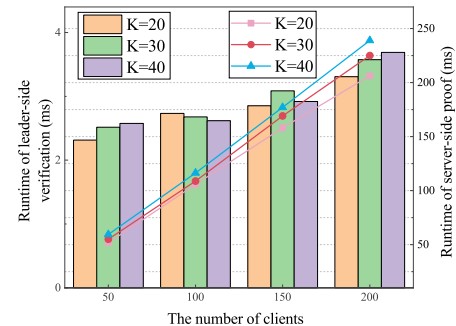

(d) Run time of verification and proof generation.

Figure 3: Runtime breakdown of SVAFD.

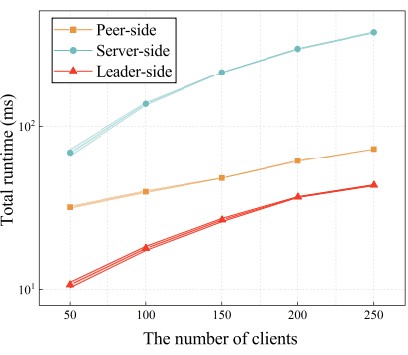

Figure 4: The overall runtime of each party.

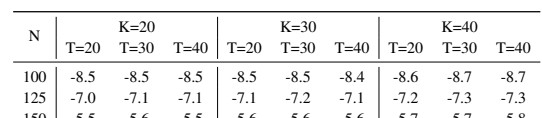

Table 1: Relative error RE (log10) for sample-grained FD.

| N | K=20 | | | K=30 | | | K=40 | | |
|---|---|---|---|---|---|---|---|---|---|
| | T=20 | T=30 | T=40 | T=20 | T=30 | T=40 | T=20 | T=30 | T=40 |
| 100 | -8.5 | -8.5 | -8.5 | -8.5 | -8.5 | -8.4 | -8.6 | -8.7 | -8.7 |
| 125 | -7.0 | -7.1 | -7.1 | -7.1 | -7.2 | -7.1 | -7.2 | -7.3 | -7.3 |
| 150 | -5.5 | -5.6 | -5.5 | -5.6 | -5.6 | -5.6 | -5.7 | -5.7 | -5.8 |

Table 2: Relative error RE (log10) for class-grained FD.

| N | K=20 | | | K=30 | | | K=40 | | |
|---|---|---|---|---|---|---|---|---|---|
| | T=20 | T=30 | T=40 | T=20 | T=30 | T=40 | T=20 | T=30 | T=40 |
| 100 | -8.0 | -7.4 | -8.1 | -7.4 | -8.0 | -8.0 | -8.0 | -8.0 | -8.1 |
| 125 | -6.1 | -6.6 | -6.3 | -6.4 | -6.2 | -6.6 | -6.2 | -6.5 | -6.6 |
| 150 | -4.9 | -4.5 | -5.1 | -4.3 | -5.1 | -4.7 | -4.9 | -4.7 | -5.1 |

verification is unrealistic. However, SVAFD addresses this by offloading the heavy signature aggregation task to the server. Finally, Fig. 3 illustrates that the server incurs an overhead $8.01\times$ and $4.84\times$ higher than the client-side (leader and peer, respectively), yet remains below $300$ ms. Overall, SVAFD effectively balances system resources, achieving secure aggregation with acceptable overhead for all parties.

**Accuracy Loss by LCC.** In this section, we evaluate the error loss introduced by LCC using the Relative Error (RE) $\frac{\|\hat{Y} - Y^*\|}{\|Y^*\|}$ with $\sigma = 10^3$, $\theta = 6$, $bs = 32$, and $\beta = 1.15$.

Table 1 and Table 2 report the results for sample-grained and class-grained FD, averaged over five runs. The relative error exhibits little sensitivity to changes in $K$ and $T$. Overall, the error at the sample granularity is consistently lower than that at the class granularity. Although the cumulative

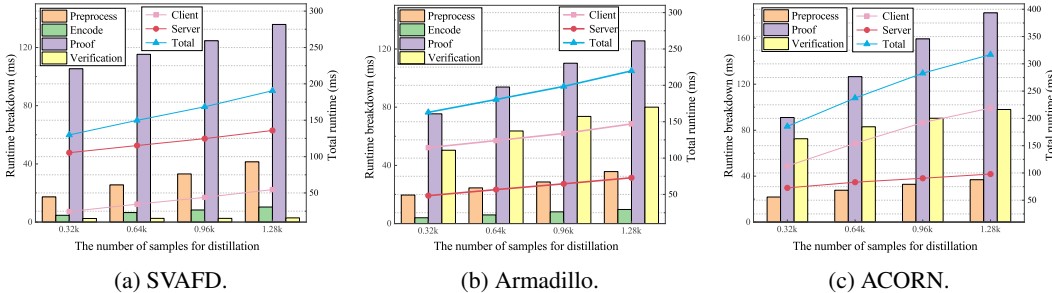

(a) SVAFD.  (b) Armadillo.  (c) ACORN.

Figure 5: Computational time per client in SVAFD, Armadillo, and ACORN Bell et al. (2023) under different numbers of distilled samples. For SVAFD, the total time includes preprocess, encoding, proof, and verification (see Appendix L for the mapping between stages and these components). Armadillo Ma et al. (2024) is decomposed into the same four parts. For ACORN Bell et al. (2023), the reported total time per client covers preprocess, proof and verification; following (Ma et al., 2024), we omit the computational and encoding costs. Note that the measured latency does not include the input-checking procedures in Armadillo Ma et al. (2024) and ACORN (Bell et al., 2023), which are detailed in Appendix M.

error from encoding operations increases with the number of clients $N$, it consistently remains on the order of $10^{-4}$. These results indicate that, with sufficient encodings, SVAFD incurs negligible decoding error and achieves high precision.

## 4.2 COMPARISON WITH STATE-OF-THE-ART

Fig. 5 reports the computational time per client, broken down into four components[6]: *Preprocess* denotes the initialization of local logits and auxiliary information; *Encode* refers to the computation of the encoded input vector; *Proof generation* is the time to construct the proof; and *Verification* measures the cost of verifying the correctness of the aggregation. In addition, we provide line plots to compare the latency across different party. See Appendix M for the full configuration, rationale, and ablations.

Across all scales, SVAFD consistently achieves the lowest total latency, followed by Armadillo (Ma et al., 2024), while ACORN Bell et al. (2023) incurs the highest cost. With 1.28k samples, the total latency of SVAFD is reduced by approximately $1.24\times$ and $1.40\times$ compared to Armadillo Ma et al. (2024) and ACORN (Bell et al., 2023), respectively. Furthermore, the total server-side latency of SVAFD exceeds the client-side latency, whereas both Armadillo Ma et al. (2024) and ACORN Bell et al. (2023) impose substantially higher costs on clients. Notably, the proportion of client computation time in SVAFD decreases by $1.58\times$. This demonstrates that SVAFD effectively shifts computation-intensive tasks to the more resourceful server side, thereby alleviating the computational and energy burden on client devices.

## 5 CONCLUSION

We present **SVAFD**, a hybrid federated distillation framework that offloads privacy-sensitive encoding and partial aggregation to clients while centralizing global decoding and proof generation on the server—transcending the centralized vs. fully decentralized dichotomy to balance privacy and efficiency. The protocol offers formal guarantees for privacy, dropout recovery, and verifiability, completing each aggregation in two rounds. Empirically, SVAFD lowers end-to-end latency versus Armadillo and ACORN by $\sim$1.24$\times$ and $\sim$1.40$\times$ under identical settings, shifts more computation to the server (client share $\downarrow$ $\sim$1.58$\times$), and achieves $\sim$0.76$\times$ Armadillo's total compute while reducing the client-side fraction by $\sim$2.37$\times$, validating secure and efficient performance for heterogeneous federated distillation.

---

[6]For comparability with other baselines, the execution stages of SVAFD are grouped into four categories, see Appendix L for the detailed mapping.

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

## A ADDITIONAL MATERIALS OF FEDERATED DISTILLATION

### A.1 OVERVIEW OF FEDERATED DISTILLATION

We investigate the federated distillation architecture, where distributed devices (called clients) collaborate to train a $D$-class classification model coordinated by the server. Without losing generality, we consider the FD system with clients set $\mathcal{C}$. Each client $c \in \mathcal{C}$ holds a private dataset $\mathcal{S}_c := \bigcup_{i=1}^{|\mathcal{S}_c|} (X_i^c, y_i^c)$ with $X_i^c$ and $y_i^c$ respectively indicating the feature vector and the corresponding label of the $i$-th training sample. Different from the homogeneous model settings of conventional FL system (Kabir et al., 2024; Lycklama et al., 2023; Rathee et al., 2023; Li et al., 2024b), we

expect that each client $c$ maintains a personalized model $\mathcal{M}^c := (\Phi^c, \ell^c)$, which differs in model parameters $\Phi^c$ and architectures. $\ell^c(\cdot)$ is the non-linear mapping determined by model $\mathcal{M}^c$. The optimization objective of the heterogeneous FD system is to maximize the average User Accuracy (UA) Wu et al. (2024b; 2023b); Mills et al. (2021) across all clients, that is to achieve generally satisfactory performance on each client.

In contrast to sharing FL model parameters or updates, FD employs the knowledge guided mechanism that is communication-efficient and heterogeneity-friendly to enable collaborative training. The server aggregates local knowledge from multiple clients to construct global knowledge, which is subsequently distributed back to the clients to enhance local training. Throughout this paper, local knowledge is defined as equivalent to the logits $\boldsymbol{\vartheta}_c$, representing the raw, unnormalized predictive confidence generated by the local model $M^c$ of client $c$ on the proxy dataset.

## A.2 DETAILS OF TWO TYPES OF FEDERATED DISTILLATION

Depending on the granularity of logits exchanged in each round of Client-Server (C/S) interactions, existing FD architectures can be classified into two categories (Wu et al., 2024b; 2023c): the Class-grained Logits Interaction-based FD Architecture (CLIA) Jeong et al. (2018b) with logits $\boldsymbol{\vartheta}_c \in \mathbb{R}^{D \times D}$ capturing class-level predictive capabilities, and the Sample-grained Logits Interaction-based FD Architecture (SLIA) Wu et al. (2024b); Li & Wang (2019); Itahara et al. (2023), where logits $\boldsymbol{\vartheta}_c \in \mathbb{R}^{O \times D}$ represent sample-level predictions, with $O$ denoting the number of distilled samples drawn from the the local dataset $\mathcal{S}_c$ (Wu et al., 2023c). SLIA enables finer-grained logits exchange.

**Class-grained Logits Interaction-based Architecture (CLIA)** For CLIA, the output of each sample $x_i^c \in \mathcal{S}_c$ for client $c \in \mathcal{C}$ needs to be close to the teacher knowledge with the same label (Jeong et al., 2018b), that is:

$$\underset{\Phi^c}{\arg\min} \sum_{(x_i^c, y_i^c) \in \mathcal{S}_c} [L_{CE}(\varkappa_0(\ell^c(X_i^c)), y_i^c) \\ + \lambda_c \cdot L_{KD}(\varkappa_0(\ell^c(X_i^c)), \varkappa_0(\vartheta^g(y_i^c)))] \tag{14}$$

where $\varkappa_0(\cdot)$ is the softmax mapping function, and $L_{CE}(\cdot)$ represents the cross-entropy loss. $Y^c(y_i^c)$ refers to the teacher knowledge from the server with the label $y_i^c$. $\boldsymbol{\vartheta}_c \in \mathbb{R}^{D \times D}$ represents the local average logits of all samples with the same label for client $c$, computed as:

$$\boldsymbol{\vartheta}_c(y_d) = Avg[\sum_{(X_i^c, y_i^c) \in \mathcal{D}^\vee \wedge y_d = y_i^c} \ell^c(X_i^c)] \tag{15}$$

CLIA supports model heterogeneity through lightweight communication. After each round of local training, the client uploads the average class-grained knowledge $\boldsymbol{\vartheta}_c$ of its local data. This is aggregated by the server to obtain teacher knowledge for the next round of local training.

**Sample-grained Logits Interaction-based Architecture (SLIA)** For SLIA, it typically requires the introduction of a public dataset or an unlabeled dataset (Li & Wang, 2019; Itahara et al., 2023). For client $c$, the logits learns from the average logits of all clients on a given sample $(X_i^o, y_i^o)$ from the public dataset $S_o$, i.e.:

$$\underset{W^c}{\arg\min} \sum_{(X_i^o, y_i^o) \in \mathcal{S}^o} L_{CE}(\varkappa_0(\ell^c(X_i^o)), \\ \varkappa_0(\vartheta^g(i))) \tag{16}$$

where, $\boldsymbol{\vartheta}^g(i)$ represents the teacher knowledge from the public accessible dataset with index $i$. At this point, the teacher knowledge $\boldsymbol{\vartheta}^g \in \mathbb{R}^{O \times D}$, where $O$ is the total number of samples selected from the public dataset. The client's local knowledge $\boldsymbol{\vartheta}_c \in \mathbb{R}^{O \times D}$ is expressed as:

$$\boldsymbol{\vartheta}_c = \underset{(X_i^o, y_i^o) \in \mathcal{S}^o}{E} \frac{\ell^c(X_i^c)}{U} \tag{17}$$

where $U$ is a hyperparameter for distribution control of the aggregated logits.

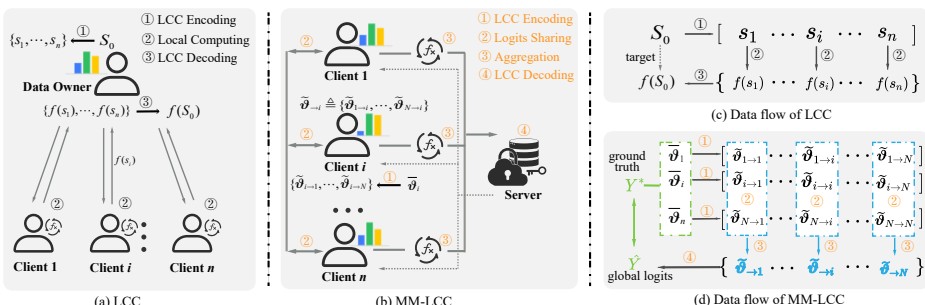

Figure 6: Workflow of LCC and MM-LCC.

## B  COMMUNICATION ROUND

**Round abstraction**  A (complete) *communication round* is the smallest unit in which (i) a message is delivered from a sender to its intended recipient(s), and (ii) the corresponding response is returned to the original sender, or a timeout is reached. This abstraction covers both client–server and server-relayed client–client exchanges.

**Timeline and time window.**  Each round $r$ starts at time $t_r$ when the server issues an Init message containing the round identifier $\text{rid} = r$ and the set of planned exchanges. The server then opens a fixed window $Win_r = [t_r, \, t_r + \Delta_r]$ and advances to the next round after $t_r + \Delta_r$. A message (or response) received after $t_r + \Delta_r$ is treated as late and is processed in the next round.

As illustrated in Fig. 6, the data provider (DP) holds a dataset $S_0$, on which a target function $f(S_0)$ is to be computed, where $f(\cdot)$ represents either linear transformations or complex polynomial operations. To enable privacy-preserving and distributed computation, the DP first encodes $S_0$ into $\mathcal{N}$ shares using Lagrange polynomial (Yu et al., 2019), and distributes them to a set of $\mathcal{N}$ service processors (SPs), i.e., the clients. Each SP performs local computation $f(\cdot)$ on its assigned share $s_n$ and returns the partial result $f(s_n)$. Finally, the DP decodes the collected results to reconstruct the global output $f(S_0)$.

The fundamental $D$-Resilient and $T$-Private property of LCC is outlined in Theorem IV of (Yu et al., 2019). $D$-Resilient indicates that even with up to $D$ clients dropping out, the provider can decode the global output $f(S_0)$. $T$-Private ensures that, even if up to $T$ clients collude, they cannot infer any meaningful information about $S_0$.

## C  LAGRANGE CODING COMPUTATION

The $(D, T)$-achievable property aligns closely with the requirements of FD for dropout-resilience and privacy-preserving, but fail to be directly applied in FD setting, where there are multiple logits data providers. Additionally, traditional LCC does not ensure the verifiability of the collaboration. To this end, we extend LCC in §3.1 to adapt it to the FD scenario as the backbone of our co-aggregation protocol. A detailed discussion of LCC's limitations for FD is provided in Appendix D.

## D  LIMITATIONS OF VANILLA LCC IN FD

**Classical LCC model.**  Classical LCC operates in a one-to-many paradigm: a single *data provider* (DP) supplies inputs for computation, and multiple *service processor* (SPs) perform coded sub-tasks and return results for decoding. The standard $(D, T)$-achievability guarantees that the master can recover the intended function despite up to $D$ stragglers, while keeping the DP's inputs information-theoretically private against any coalition of up to $T$ SPs.

**FD differs fundamentally.**  FD induces a multi-to-multi interaction pattern:

- **Multiple DPs and SPs.** Each client acts simultaneously as a DP (providing logits for aggregation) and as an SP (offering local aggregation). Hence, there are multiple, concurrent providers and multiple sinks.

- **Lack of verifiability.** Vanilla LCC provides $(D, T)$ tolerance and input privacy but does not natively ensure *verifiable* collaboration (e.g., correctness of coded partial results).

**Our remedy.** § 3.1 develops an FD-tailored extension of LCC (MM-LCC). In this design, the encoding and partial aggregation tasks are delegated to the clients, while the decoding and the additional signature aggregation tasks are offloaded to the server. This division preserves client-side data privacy while balancing the computational load across participants.

## E  CLASS-GRAINED SPLIT

Prior to applying LCC encoding, the logits $\boldsymbol{\vartheta}_i$ of client $i \in \mathcal{N}_c$ are first partitioned into $K$ splits. Besides the sample-grained, We consider another form of logits splitting methods for class-grained FD architecture.

*Class-grained Split*. In the class-grain FD, the logits mode of client $i \in \mathcal{N}_c$ is only related to the class numbers $D$, namely $\vartheta_i \in \mathbb{R}^{D \times D}$.

With additive splitting method, we split $\boldsymbol{\vartheta}_i \in \mathbb{R}^{D \times D}$ into $K$ parts to obtain $\overline{\boldsymbol{\vartheta}}_i \triangleq [\overline{\vartheta}_i^{(1)}, \overline{\vartheta}_i^{(2)}, \ldots, \overline{\vartheta}_i^{(K)}] \in \mathbb{R}^{K \times D \times D}$, where each split $\overline{\vartheta}_i^{(k)} \in \mathbb{R}^{D \times D}$ for $k \in [K]$. Let $v_{g,l}$ denote the $(g, l)$-th element of $\overline{\boldsymbol{\vartheta}}_i$, and $v_{g,l}^{(k)}$ the corresponding element in the $k$-th split $\overline{\vartheta}_i^{(k)}$. The additive property satisfies $v_{g,l} = \sum_{k=1}^{K} v_{g,l}^{(k)}, \quad \forall g, l \in [D]$. This split method is also applicable to sample-grain FD, but for efficiency, the following method is proposed (Liu et al., 2023).

## F  MASKING (BLINDING) MATRIX

**Definition.** Fix a block shape $\Omega \times D$ as in the main text, where each decoded block $\overline{Y}_i^{(k)} \in \mathbb{R}^{\Omega \times D}$ (class-grained takes $\Omega = D$; sample-grained uses the per-block row count). For each leader $c$, draw a *masking matrix* $\mathbf{R}_c \in \mathbb{R}^{\Omega \times D}$ with entrywise nonzeros, i.e., $(\mathbf{R}_c)_{u,v} \neq 0$ for all $(u, v)$, over a ground field $\mathbb{R}$. Define the entrywise (Hadamard) masking operator $\mathsf{M}_{\mathbf{R}_c}(X) = \mathbf{R}_c \odot X$ and its inverse $\mathsf{M}_{\mathbf{R}_c}^{-1}(X) = \mathbf{R}_c^{-1} \odot X$, where $\mathbf{R}_c^{-1}$ is the entrywise reciprocal. Then $\mathsf{M}_{\mathbf{R}_c}^{-1}(\mathsf{M}_{\mathbf{R}_c}(X)) = X$ for all conformable $X$.

**Blinding personalized weights.** In a given round, let $\mathcal{W}_c = [w_i]_{i \in \mathcal{N}_c}$ be the leader's custom aggregation weights (e.g., $\sum_i w_i = 1$). To conceal $\mathcal{W}_c$ from the server and peers while preserving linearity, leader $c$ derives *masked weights* by broadcasting $\mathbf{R}_c$ onto each scalar:

$$\widetilde{W}_{c,i} := w_i \mathbf{R}_c \in \mathbb{R}^{\Omega \times D}, \qquad i \in \mathcal{N}_c.$$

Stacking these yields $\widetilde{\mathcal{W}}_c = \left[\widetilde{W}_{c,i}\right]_{i \in \mathcal{N}_c} \in \mathbb{R}^{N \times \Omega \times D}$. Leader $c$ distributes to each follower $i \in \mathcal{N}_c$ its own $\widetilde{\mathcal{W}}_c$ together with the initialization data (Lagrange coefficient matrix). Given the masked weights, peer $i$ computes its masked partial aggregation $\hbar(\widetilde{\vartheta}_{\rightarrow i}) = \widetilde{W} f(\widetilde{\vartheta}_{\rightarrow i})$, and submits it for global decoding. By linearity of $\odot$ and the aggregator, the server obtains the masked global logits. After reassembly across blocks (Eq. equation 7), the leader removes the mask to recover the intended weighted results.

## G  DATA PRECISION CONVERSION

We take the precision conversion function as $w_i \leftarrow \mathbf{Conv}(w_i, q) = \lfloor w_i * 10^q \rfloor$, and $V_i^{(k)} \leftarrow \mathbf{Conv}(V_i^{(k)}, q) = \lfloor V_i^{(k)} * 10^q \rfloor$. Furthermore, for each entry $y_{g,l}$ in the decoded $\hat{Y}_c$, we take $y_{g,l} \leftarrow \mathbf{Conv}(y_{g,l}, q) = \lfloor y_{g,l} * 10^{2q} \rfloor$. This approximation process inevitably results in some information loss and a reduction in precision, however, it does not impact the accuracy of the verification.

# H  BILINEAR PAIRINGS.

The bilinear pairing group is define as $(p, \mathbb{G}, \mathbb{G}_T, e)$, where $p$ is a large prime number, $\mathbb{G}$ and $\mathbb{G}_T$ are prime-order cyclic multiplicative groups with order $p$, and $e$ is a bilinear map function as $e : \mathbb{G} \times \mathbb{G} \to \mathbb{G}_T$. The bilinear map satisfies the following three properties:

- **Bilinearity**: For all $a, b \in \mathbb{Z}_p$ and $g \in \mathbb{G}$, $e(g^a, g^b) = e(g, g)^{ab}$.

- **Non-degeneracy**: There exists an element $g \in \mathbb{G}$ such that $e(g, g) \neq 1$.

- **Computability**: For all $g \in \mathbb{G}$, there is an efficient algorithm to compute $e(g, g)$.

# I  PROOF OF THE LEMMA 3.2

While the goal of group leader $c$ is to aggregate the logits from its $N$ followers within the peer set $\mathcal{N}_c$, that is, to obtain the ground-truth global aggregated logits $Y_c^* = \mathcal{W}_c f(\overline{\mathcal{V}}^c)$, with $\overline{\mathcal{V}}^c =$

SVAFD enables leader $c$ to acquire a decoded global logits $\hat{Y}_c = \frac{1}{\mathbf{R}_c} \odot Dec(\hbar(\widetilde{\mathcal{V}}^c))$, where $\widetilde{\mathcal{V}}^c = \{\widetilde{\vartheta}_{\to i} | i \in \mathcal{N}_c\}$. The correctness of co-aggregation can be formally proven as:

$$
\begin{aligned}
\hat{Y}_c &= \frac{1}{\mathbf{R_c}} \odot \mathrm{Dec}\left(\hbar(\widetilde{\mathcal{V}}^c)\right) \\
&\overset{(1)}{=} \frac{1}{\mathbf{R_c}} \odot \mathrm{Dec}\left(\widetilde{\mathcal{W}}_c f(\mathrm{Enc}(\overline{\mathcal{V}}^c))\right) \\
&\overset{(2)}{=} \frac{1}{\mathbf{R_c}} \odot \mathbf{R_c} \odot \mathcal{W}_c f(\mathrm{Dec}(\mathrm{Enc}(\overline{\mathcal{V}}^c))) \overset{?}{=} Y_c^*
\end{aligned}
\tag{18}
$$

As the data flow diagram illustrated, each peer $i \in \mathcal{N}_c$ first splits the local knowledge $\boldsymbol{\vartheta}_i$ to get $\overline{\boldsymbol{\vartheta}}_i = [\overline{\vartheta}_i^{(1)}, \overline{\vartheta}_i^{(2)}, \dots, \overline{\vartheta}_i^{(K)}]$, performs LCC encoding operation $Enc(\cdot)$ for logits shares $\widetilde{\boldsymbol{\vartheta}}_{i\to} = [\widetilde{\vartheta}_{i\to 1}, \widetilde{\vartheta}_{i\to 2}, \cdots, \widetilde{\vartheta}_{i\to N}]$ and exchanges these shares with other peers. The collection of all received shares forms the encoded logits set $\widetilde{\mathcal{V}}^c = \{\widetilde{\vartheta}_{\to i} | i \in \mathcal{N}_c\}$, and the above process can be simplified as $\widetilde{\mathcal{V}}^c = \mathrm{Enc}(\overline{\mathcal{V}}^c)$, where $\overline{\mathcal{V}}^c = [\overline{\vartheta}_1, \overline{\vartheta}_2, \cdots, \overline{\vartheta}_N] \Leftrightarrow \{\overline{\mathcal{V}}^{(k)} | k \in [K]\}$. Combining with Eq. 2, we can deduce the equation '(1)' in Eq. 18 holds

According to the Lagrange encoding function in Eq. 4, $u_i(\alpha_j), j \in \mathcal{N}_c$ is a polynomial of maximum degree $(K + T - 1)$ with respect to $\alpha$, satisfying that $u_i(\alpha_j) = \widetilde{\vartheta}_{i\to j}$ and $u_i(\beta_k) = \overline{\vartheta}_i^{(k)}, k \in [K]$. By defining the set $u(\cdot) = [u_1(\cdot); u_2(\cdot); \cdots; u_{n'}(\cdot)]$, we have:

$$
u(\Xi) = \begin{cases} \widetilde{\boldsymbol{\vartheta}}_{\to \mathcal{I}(\Xi)}, & \Xi \in \{\alpha_x | x \in \mathcal{N}_c\} \\ \overline{\mathcal{V}}^{(\Xi)}, & \Xi \in \{\beta_k | k \in [K]\} \end{cases}
\tag{19}
$$

where $\mathcal{I}(\Xi)$ denotes the subscript of $\Xi$. Since $\hbar(\widetilde{\boldsymbol{\vartheta}}_{\to i}) = \hbar(u(\alpha_i)) = \widetilde{\mathcal{W}}_c f(u(\alpha_i))$, which poses a degree of $deg(f \times u) = D \times (K + T - 1)$. When the aggregation results $\hbar(\widetilde{\mathcal{V}}^c) = \{\hbar(\widetilde{\boldsymbol{\vartheta}}_{\to i}) | i \in \mathcal{N}_c\}$ are submitted to the central server, at least $D \times (K + T - 1) + 1$ values are required to reconstruct the polynomial $\hbar(u(\cdot))$ via Lagrange polynomial interpolation, e.g. $|\mathcal{N}_c'| > D \times (K + T - 1) + 1$, where $\mathcal{N}_c' \subset \mathcal{N}_c$ denotes the set of clients in $\mathcal{N}_c$ that remain after excluding stragglers. Then $\widetilde{Y}_c$ can be induced by evaluating $\hbar(u(\beta_k)), k \in [K]$. The above decoding process can be expressed as $\mathrm{Dec}(\hbar(\widetilde{\mathcal{V}}^c)) = \hbar(\{\overline{\vartheta}^{(k)} | k \in [K]\}) = \widetilde{\mathcal{W}}_c f(\{\overline{\vartheta}^{(k)} | k \in [K]\}) = R_c \odot \mathcal{W}_c f(\overline{\mathcal{V}}^c)$. Since $\mathrm{Dec}(\mathrm{Enc}(x)) = x$, the equation '(2)' in Eq.18 is holds, thus the $\hbar$-operational of Theorem 3.1 and the Lemma 3.2 Homomorphic Share Aggregation are proved.

# J  PROOF FOR THE $I$-VERIFIABLE PROPERTY OF THEOREM 3.1

This section provides the verification of the linear weighted aggregation for FD, which is based on the Verifiable Linear Computing process of (Liu et al., 2023). The formal proof is given by the

following equation:

$$\Pi_c = \prod_{i \in \mathcal{N}_c} e(\prod_{k=1}^{K} \pi_i^{(k)'}, \pi_i^{''}) = \prod_{i \in \mathcal{N}_c} e(g^{\sum_{k=1}^{K}(V_i^{(k)} + \Upsilon_i)}, g^{w_i})$$

$$= \prod_{i \in \mathcal{N}_c} e(g, g)^{(\sum_{k=1}^{K} \sum_{g=1}^{\Omega} \sum_{l=1}^{D} v_{i,gl}^{(k)} w_i + K \Upsilon_i w_i)} \tag{20}$$

$$= e(g, g)^{\sum_{i \in \mathcal{N}_c} w_i \sum_{k=1}^{K} \sum_{g=1}^{\Omega} \sum_{l=1}^{D} v_{i,gl}^{(k)}} e(g, g)^{K \sum_{i \in \mathcal{N}_c} w_i \Upsilon_i}$$

$$= e(g, g)^{\sum_{g=1}^{M} \sum_{l=1}^{D} y_{g,l} + K \sum_{i \in \mathcal{N}_c} w_i \Upsilon_i}$$

where $y_{g,l}$ represents the $(g, l)$-th element of the teacher's knowledge $\hat{Y}_c$, $v_{i,gl}^{(k)}$ denotes the $(g, l)$-th element of $\overline{\vartheta}_i^{(k)}$. Any clients or the server, who has dishonest behavior during computation and is not detected, needs to make equation $\Pi_c \overset{?}{=} e(g, g)^{\sum_{g=1}^{M} \sum_{l=1}^{D} y_{g,l} + K \sum_{i \in \mathcal{N}_c} w_i \Upsilon_i}$ hold. However, for the peers, it receives the masked $\widetilde{\mathcal{W}}_c$ and does not know the decoded value of $\hat{Y}_c$ or the private keys $\Upsilon_i, i \in \mathcal{N}_c$ of other participants. Similarly, the server only poses the masked $\widetilde{Y}_c$. Therefore, even in the worst-case scenario where the server colludes with up to $N - 1$ clients, they still cannot forge a valid verification proof $\Pi_c$ to mislead the leader into accepting tampered global logit. The $I$-Verifiable property of SVAFD described in Theorem 3.1 is proved.

# K    PROOF FOR THE $T$-PRIVACY PROPERTY OF THEOREM 3.1

For each peer set $\mathcal{N}_c, c \in \mathcal{N}$, there are three types of identities: group leader $c$, following peers $i \in \mathcal{N}_c$, and the server. Sensitive data include auxiliary information and knowledge information. The former indicates the signatures $(\pi_{\mathcal{W}}^c, \pi_{\mathcal{V}}^c)$, while the later denotes the local logits $\overline{\vartheta}_i \triangleq [\overline{\vartheta}_i^{(1)}, \overline{\vartheta}_i^{(2)}, \dots, \overline{\vartheta}_i^{(K)}]$. The proposed SVAFD scheme effectively protects the privacy of these two types of data. It is worth noting that we assume the decoded global knowledge is not sensitive, as it comes from the aggregation of multiple student knowledge, rather than any single party.

### K.0.1    PRIVACY PROTECTION OF AUXILIARY INFORMATION

The security of $(\pi_{\mathcal{W}}^c, \pi_{\mathcal{V}}^c)$ relies on the discrete logarithm problem (DLP). In the multiplicative group $\mathbb{G}$, it is computationally straightforward to get $y = g^a \mod p$ for an integer $a \in \mathbb{Z}_p$. However, given the parameters $(\mathbb{G}, p, g, y)$, no Probabilistic Polynomial Time (PPT) algorithm $\mathcal{A}$ should be able to efficiently solve DLP, as expressed by the following advantage equation, for any polynomial $p(\cdot)$ and sufficiently large $\lambda$.

$$\Pr[\mathcal{A}(g, y, p) = a \wedge (g^a \mod p = y)] < \frac{1}{p(\lambda)}. \tag{21}$$

### K.0.2    PRIVACY PROTECTION OF LOGITS INFORMATION

To prove the privacy protection of knowledge information, three simulators need to be constructed.

**1) C-C:Client's privacy (to client):** For each followed client $i$ in group $\mathcal{N}_c$, no other follower $j \in \mathcal{N}_c$ can obtain any private knowledge information $\overline{\vartheta}_i$ from the received logits share $\widetilde{\vartheta}_{i \to j}$.

Proof: The view of client $j$ regarding $i$ is given as

$$View_{CC} = \{\widetilde{\vartheta}_{i \to j}, \mathcal{A}_c \cup \mathcal{B}_c, \{\widetilde{\vartheta}_j^{(k)}\}_{k=1}^{K}, \{\widetilde{h}_j^{(t-K)}\}_{t=K+1}^{K+T}\} \tag{22}$$

where $\mathcal{A}_c = \{\alpha_i, i \in \mathcal{N}_c\}$ and $\mathcal{B}_c = \{\beta_j, j \in [K + T]\}$. The simulator $S_{cc}$ takes as input $(\widetilde{\vartheta}_{i \to j}, \mathcal{A}_c \cup \mathcal{B}_c)$ and randomly generates pseudo-noise slices and pseudo-knowledge slices as $\{\widetilde{\vartheta}_j^{'(k)}\}_{k=1}^{K}$ and $\{\widetilde{h}_j^{'(t-K)}\}_{t=K+1}^{K+T}$, that satisfying $\widetilde{\vartheta}_{i \to j}' = \sum_{k=1}^{K} \overline{\vartheta}_j^{(k)} l_k(\alpha_j) +$

$\sum_{t=K+1}^{K+T} \overline{h}'^{(t-K)}_j l_t(\alpha_j) = \widetilde{\vartheta}_{i \to j}$. Thus we can obtain

$$S_{cc}(\widetilde{\vartheta}'_{i \to j}, \mathcal{A}_c \cup \mathcal{B}_c) = \{\widetilde{\vartheta}'_{i \to j}, \mathcal{A}_c \cup \mathcal{B}_c, \\ \{\widetilde{\vartheta}^{(k)}_j\}_{k=1}^K, \{\widetilde{h}^{(t-K)}_j\}_{t=K+1}^{K+T}\}. \tag{23}$$

Since $\widetilde{\vartheta}'_{i \to j} \overset{c}{\equiv} \widetilde{\vartheta}_{i \to j}$, therefore $View_{cc}(\widetilde{\boldsymbol{\vartheta}}_i) \overset{c}{\equiv} S_{cc}(\widetilde{\vartheta}_{i \to j}, \mathcal{A}_c \cup \mathcal{B}_c)$.

**2) C-S:Client's privacy(to server)** For each followed client $i$ in group $\mathcal{N}_c$, the server obtains nothing about the private knowledge information $\widetilde{\boldsymbol{\vartheta}}_i$ from its own view.

Proof: The server's view is represented as $View_{cc} = \{\hbar(\widetilde{\mathcal{V}}^c), \widetilde{Y}_c, \mathcal{A}_c \cup \mathcal{B}_c\}$, with $\hbar(\widetilde{\mathcal{V}}^c) = \{\hbar(\widetilde{\boldsymbol{\vartheta}}_{\to i}) \mid i \in \mathcal{N}_c\}$, and its output being the decoded teacher knowledge $\widetilde{Y}_c$. Therefore, the input of simulator $Scs$ is $\{\hbar(\overline{\mathcal{V}}^c), \widetilde{Y}_c, \mathcal{A}_c \cup \mathcal{B}_c\}$, which will construct the Lagrange coefficient matrix for decoding based on $\mathcal{A}_c \cup \mathcal{B}_c$, and generate the pseudo-knowledge data $\widetilde{\mathcal{V}}'_c$ and the weight matrix $\widetilde{\mathcal{W}}'$ to satisfy $\widetilde{\mathcal{W}}' f(\widetilde{\mathcal{V}}'_c) = \hbar(\widetilde{\mathcal{V}}^c)$, which is decoded to obtain the pseudo-teacher knowledge $\widetilde{Y}'_c$. Thus we can obtain

$$S_{cs}(\hbar(\overline{\mathcal{V}}^c), \widetilde{Y}_c, \mathcal{A}_c \cup \mathcal{B}_c) = \{\hbar'(\widetilde{\mathcal{V}}'_c), \widetilde{Y}'_c, \mathcal{A}_c \cup \mathcal{B}_c\}. \tag{24}$$

Since $\hbar'(\widetilde{\mathcal{V}}^c) = \widetilde{\mathcal{W}}' f(\widetilde{\mathcal{V}}'_c) = \hbar(\widetilde{\mathcal{V}}^c)$, then $\hbar'(\widetilde{\mathcal{V}}^c) \overset{c}{\equiv} \hbar(\widetilde{\mathcal{V}}'_c)$, and after decoding, it holds $\widetilde{Y}'_c \overset{c}{\equiv} \widetilde{Y}_c$. Therefore, $View_{cs} \overset{c}{\equiv} S_{cs}(\hbar(\overline{\mathcal{V}}^c), Y_c, \mathcal{A}_c \cup \mathcal{B}_c)$.

**3) C-SC:Client's privacy to Server Collude with most T-clients** In the group $\mathcal{N}_c$, the mutual information leakage $\eta_c$ between the collusion view of the server with T clients, and the local view of client $i \in \mathcal{N}_c$ satisfies $\eta_c \leq \delta$, where $\delta$ is the upper bound of the mutual information, describing the tolerable threshold of privacy leakage.

Proof: Let $N_T = \{i_0, i_1, \ldots, i_t\}$ represent the index set of colluding parties, where $i_0$ denotes the server. The colluding view is the set of encoded slices they receive $View_{cs} = L_T^1 V_i + L_T^2 H_i$, where $L_T^1 \in \mathbb{C}^{R \times K}$ and $L_T^2 \in \mathbb{C}^{R \times T}$ are divided into two parts by the Lagrange multiplier $L_T \subset L_c$ along the columns. We utilize $v_{g,l}^{(k)}$ and $h_{g,l}^{(t)}$ represent the $(g, l)$-th element of $\overline{\vartheta}_i^{(k)}$ and $\overline{h}_i^{(t)}$, respectively. $V_i = [v_{g,l}^{(1)}, \cdots, v_{g,l}^{(K)}] \in \mathbb{R}^K$ and $H_i = [h_{h,l}^{(1)}, \cdots, h_{g,l}^{(t)}] \in \mathbb{R}^T$. The local view of client $i$ is the splited knowledge $View_C = V_i$. Then the amount of information leakage $\eta_c$ by the collusion set $N_T$ can be measured by the MIS metric according to Theorem.6 in Soleymani et al. (2021) as :

$$\eta_c \overset{\text{def}}{=} \max_{N_T} \max_{\substack{P_V : |v_{g,l}^{(k)}| \leq r, \forall k \in [K] \\ g \in \Omega, \ l \in D}} I(View_{cs}; View_c) \\ \leq \max_{N_T} \max_{\substack{P_V : E[v_{g,l}^{(k)2}] \leq r^2, \forall k \in [K] \\ g \in \Omega, \ l \in D}} I(V_i; L_T^1 V_i + L_T^2 H_i). \tag{25}$$

where $P_V$ is the probability density function (PDF) of $V_i$, and the maximization is performed over all $N_T \subset \mathcal{N}_c$ such that $|N_T| = T$. Since $|v_{g,l}^{(k)}| \leq r$, we scale this inequation by constraining the expected value of the square $E[v_{g,l}^{(k)}]^2 \leq r^2$ instead of its absolute value.

Consider a MIMO channel with $K$ transmit and $T$ receive antennas and the input-output relation $y = Lx + H$. Here $L$ represents the channel gain matrix, and $H$ represents the Gaussian noise vector, which are equivalent to $L_T^1$ and $L_T^2 H_i$ in $View_{cs}$, respectively. Then, Under equal-power allocation constraint, the maximization of $I(V_i; L_T^1 V_i + L_T^2 H_i)$ is equivalent to the capacity of the MIMO channel $C = \log_2 |I_t + PN_c^{-1} LL^H|$, where, $P$ is the maximum transmission power of each antenna at the transmitter, $I_t$ is the $t \times t$ identity matrix, $|\cdot|$ denotes the matrix determinant, and $N_c$ denotes the covariance matrix of $H$. Then, an upper bound on the right-hand side of $\eta_c$ is concluded as shown in Theorem.6 of Soleymani et al. (2021) as:

$$\eta_c \leq \max_{N_T} \log_2 \left| I_t + \frac{r^2 t}{\sigma_{trun}^2} \tilde{\Sigma}_T^{-1} \Sigma_T \right| \tag{26}$$

where $\tilde{\Sigma}_T = L_T^2 L_T^{2\,H}$, $\Sigma_T = L_T^1 L_T^{1\,H}$, and the variance of the noise after truncation denotes as follows:

$$\sigma_{\text{trunc}}^2 = \sigma^2 \left( 1 - \frac{\theta \cdot \phi(\theta) - (-\theta) \cdot \phi(-\theta)}{\Phi(\theta) - \Phi(-\theta)} \right.$$
$$\left. - \left( \frac{\phi(\theta) - \phi(-\theta)}{\Phi(\theta) - \Phi(-\theta)} \right)^2 \right) \tag{27}$$

For more proof information, please refer to the original paper (Soleymani et al., 2021).

## L  STAGE MAPPING FOR SVAFD

To enable comparisons across heterogeneous baselines (e.g., Armadillo (Ma et al., 2024), ACORN (Bell et al., 2023)), we collapse the micro-stages (as shown in Fig. 3) of our SVAFD pipeline into four canonical macro-categories that are reported in Fig. 5: *Preprocess*, *Encode*, *Proof*, and *Verification*. This normalization removes system-specific granularity while preserving the semantic roles of initialization.

**Preprocess.** Sum of (i) client-side initialization latency (ms) and (ii) client-side auxiliary-information generation latency (ms). These operations prepare the local state and metadata before any transform or aggregation.

**Encode.** Sum of (i) Lcc encoding and (ii) partial aggregation on the client-side. This stage captures the transformation of local logits into their encoded representation together with the client-side aggregation prescribed by the SVAFD pipeline.

**Proof.** Sum of (i) LCC decode latency plus (ii) signature aggregation time. In our stack these two server-side procedures jointly produce the correctness artefact that is subsequently consumed by the client, and thus play the role analogous to the prover's construction in cryptographic baselines.

**Verification.** Client verification latency (ms). This final step performs correctness checks and de-masking on the client side and is the natural counterpart of the verifier's work in the baseline literature.

## M  BASELINE CONFIGURATIONS AND ADDITIONAL EXPERIMENTAL DETAILS (ARMADILLO MA ET AL. (2024) AND ACORN BELL ET AL. (2023))

**Baselines and Hyperparameters.** To ensure comparability, we align all baselines with $n = 100$, corruption rate $\eta = 0.1$, and dropout rate $\delta = 0.1$. For *Armadillo* (Ma et al., 2024), we use the *GOD-only* variant (denoted Armadillo-noIV), which preserves Guaranteed Output Delivery while removing input-validity checks on $x$ (no $L_2/L_\infty$ tests for $x$), but retains the noise bound on $e$ and membership checks needed for modulus separation. For ACORN (Bell et al., 2023), we *remove* the task-level predicate proofs $\pi_{\text{valid}(x)}$ (e.g., $L_2$-bound or $k$-hot) while *retaining* (i) the encoding-validity proof $\pi_{\text{Enc}}(sk, x)$; (ii) the range/smallness proof $\pi_{0 \le x < t}$ required to preclude plaintext wrap-around, and (iii) the distributed key-correctness check for $\sum_i sk_i = sk$. Unless otherwise stated, we use $t = 2^{16}$ (FL-style 16-bit quantization), vector commitments and inner-product proofs are batch-verified.

Since both Armadillo Ma et al. (2024) and ACORN Bell et al. (2023) did not have implementation and benchmarks, We re-implement both them at the protocol level, strictly following the respective specifications (relations/constraints, inequalities, and message schedules) and the reported measurement methodology. Because the public artifacts emphasize specification and empirical trends rather than a unified reference codebase, certain implementation-level constants (e.g., curve/field backends, MSM windowing, batching granularity, memory layout) may necessarily differ across codebases. Consequently, minor deviations from previously reported micro-benchmarks are possible. All such choices are kept conventional and within the prescribed parameter regimes.

---

**Algorithm 1** Round 1: Knowledge Aggregation

---

1: **Input:** Leader $c$, Peer set $\mathcal{N}_c$, Logits sets $\mathcal{V} = [\boldsymbol{\vartheta}_1, \ldots, \boldsymbol{\vartheta}_N]$, weights set $\mathcal{W} = [w_1, \cdots, w_N]$, Split size $K$, Privacy parameters $T$, Dropout number $D$, Batch size $O$.

2: **Output:** Partial aggregated logits $\{\hbar(\widetilde{\vartheta}_{\to j}) \mid j \in \mathcal{N}_c\}$, Logits signature $\pi_{\mathcal{V}}^c$, Weights signature $\pi_{\mathcal{W}}^c$.

3: // Take the $\mathcal{N}_c$ as an example, and all the groups follow the same rules.

4: **Data Preprocess:**

5: **for** each client $i \in \mathcal{N}_c$ **do**

6:     Split $\boldsymbol{\vartheta}_i$ into $K$ shares: $\overline{\boldsymbol{\vartheta}}_i \in \mathbb{R}^{K \times \Omega \times D}$.

7:     Generate random matrix $\overline{H}_i \in \mathbb{C}^{T \times \Omega \times D}$ and blind logits $B(\overline{\boldsymbol{\vartheta}}_i)$.

8: **end for**

9: **Lagrange Matrix and Weights Blind:**

10: Leader $c$ computes Lagrange coefficient matrix.

11: Leader generates the blinded weight matrix $\widetilde{\mathcal{W}}_c = [w_1 \cdot \mathbf{R}_c, \ldots, w_N \cdot \mathbf{R}_c]$.

12: Distribute $(L_c, \widetilde{\mathcal{W}}_c)$ to followers in $\mathcal{N}_c$ during initialization.

13: **LCC Encode:**

14: **for** each peer $i \in \mathcal{N}_c$ **do**

15:     **for** each $\alpha_j, j \in \mathcal{N}_c$ **do**

16:         Perform LCC encoding $\widetilde{\vartheta}_{i \to j} = \sum_{k=1}^{K} \overline{\vartheta}_i^{(k)} l_k(\alpha_j) + \sum_{t=K+1}^{K+T} \overline{h}_i^{(t-K)} l_t(\alpha_j)$.

17:     **end for**

18:     Send $[\widetilde{\vartheta}_{i \to 1}, \ldots, \widetilde{\vartheta}_{i \to N}]$ to each peer according to the index.

19: **end for**

20: **Auxiliary Information:**

21: **for** each peer $i \in \mathcal{N}_c$ **do**

22:     Compute digest: $V_i^{(k)} = \sum_{g=1}^{\Omega} \sum_{l=1}^{D} v_{g,l}^{(k)}$.

23:     Compute auxiliary proof: $\pi_i^{(k)'} = g^{V_i^{(k)} + \Upsilon_i}, \forall k \in [K]$.

24:     Upload $\pi_i' = \{\pi_i^{(1)'}, \ldots, \pi_i^{(K)'}\}$ to the server while sent the leader $\Upsilon_i$.

25: **end for**

26: Leader signs weights: $\pi_i^c = g^{w_i}, \forall i \in \mathcal{N}_c$, and upload auxiliary proofs $\pi_{\mathcal{W}}^c$ to server.

27: **Local Aggregation:**

28: **for** each follower $i \in \mathcal{N}_c$ **do**

29:     Receive logits shares: $\widetilde{\boldsymbol{\vartheta}}_{\to i} = [\widetilde{\vartheta}_{1 \to i}, \ldots, \widetilde{\vartheta}_{N \to i}]$.

30:     Perform aggregation: $\hbar(\widetilde{\boldsymbol{\vartheta}}_{\to i}) = \widetilde{\mathcal{W}}_c f(\widetilde{\boldsymbol{\vartheta}}_{\to i})$.

31: **end for**

32: Deliver aggregated results $\{\hbar(\widetilde{\boldsymbol{\vartheta}}_{\to i}) \mid i \in \mathcal{N}_c\}$ to the server.

---

## M.1 ARMADILLO CONFIGURATION

We instantiate the Armadillo Ma et al. (2024) baseline in a *Guaranteed Output Delivery (GOD)-only* configuration, denoted Armadillo-noIV, which follows the protocol-level specification and security arguments while *removing input-validity checks on the client update $x$*. In particular, we drop the $L_2/L_\infty$ constraints and the associated zero-knowledge sub-proofs for $x$, but we *retain* (i) the linear relation $y = Ak + e + \Delta x$, (ii) the $L_2$ bound on $e$ (proven via a 256-row Johnson–Lindenstrauss projection), and (iii) the membership-style $L_\infty$ checks ensuring coordinates lie in $[0, q)$ that are required by the modulus-separation ("reverse proof modulo $q$") optimization. The inner aggregation keeps the packed Shamir consistency and binding tests (modified SCRAPE), preserving robustness and GOD.

**LWE and proof-system parameters.** We set the LWE dimension $\lambda = 1200$, ciphertext modulus $q = 2^{32}$ (machine-friendly), and plaintext scale $p = 2^{16}$. Vector commitments and inner-product proofs are implemented over a prime-order group of size $Q \approx 2^{255/256}$, with Nova (R1CS) as the proof system. We adopt the modulus-separation trick (reverse proof modulo $q$), which requires membership ($< q$) checks but *not* input-validity thresholds on $x$.

---

**Algorithm 2** Round 2: Knowledge Verification

---

1: **Input:** Aggregated partial logits $\{\hbar(\widetilde{\vartheta}_{\to i}) \mid i \in \mathcal{N}_c\}$, weights signatures $\pi_{\mathcal{W}}^c$, logits signatures $\pi_{\mathcal{V}}^c$.

2: **Output:** Decoded teacher knowledge $\hat{Y}_c$, Verification proof $\Pi_c$.

3: // Take the $\mathcal{N}_c$ as an example, and all the peer sets follow the same rules.

4: **LCC Decode:**

5: Server collects aggregated logits $\{\hbar(\widetilde{\vartheta}_{\to i}) \mid i \in \mathcal{N}_c\}$ from all the groups.

6: Interpolate polynomial $u(\alpha_i)$ using $deg(f(x))(K + T - 1) + 1$ shares.

7: **for** each $k \in [K]$ **do**

8:    Evaluate $u(\beta_k)$ to get $\overline{Y}_c^{(k)} = \hbar(\overline{\mathcal{V}}^{(k)})$.

9: **end for**

10: Decode global knowledge: $\widetilde{Y}_c = [\overline{Y}_c^{(1)}, \overline{Y}_c^{(2)}, \ldots, \overline{Y}_c^{(K)}]$.

11: **Signature Aggregation:**

12: Compute aggregated proof $\Pi_c = \prod_{i \in \mathcal{N}_c} e\left(\prod_{k=1}^K \pi_i^{(k)'}, \pi_i''\right)$.

13: Deliver $Res_c = (\widetilde{Y}_c, \Pi_c)$ to leader $c$.

14: **Proof Verification:**

15: Leader $c$ performs reverse split on $\widetilde{Y}_c$.

16: Remove blind factor $\hat{Y}_c = \frac{1}{\mathbf{R}_c} \odot \widetilde{Y}_c$.

17: Verify correctness of global logits.

18: Check if $e(g, g)^{\sum_{g=1}^M \sum_{l=1}^D y_{g,l} + K \sum_{i \in \mathcal{N}_c} w_i \Upsilon_i} \stackrel{?}{=} \Pi_c$.

---

**Helpers, slicing, and packed Shamir.** To make the helper set feasible at $n = 100$, we employ *vector slicing* with factor $s = 32$, so the number of packed secrets per instance is $a = \lceil \lambda/s \rceil = 38$. We take $C = 60$ helpers and choose the Shamir degree $d = 50$, which satisfies the protocol's feasibility inequalities under $\eta_C = \delta_C = 0.1$:

$$\text{(i)} \quad d - a > C\eta_C, \qquad \text{(ii)} \quad d < C(1 - \delta_C), \qquad \text{hence} \quad C > \frac{a}{1 - \delta_C - \eta_C}.$$

Numerically, $a/(1 - \delta_C - \eta_C) = 38/0.8 = 47.5 < C = 60$, and $(1 + \frac{3}{2}\eta_C)a = 43.7 < d = 50 < 51.3 = \frac{3}{2}a(1 - \delta_C)$. The Shamir sharing operates over a 32-bit prime field $\mathbb{F}_{\hat{q}}$, distinct from the LWE modulus $q$.

## M.2 ACORN CONFIGURATION

We adopt the ACORN-detect setting and remove task-specific validity predicates. Concretely, we drop $\pi_{\text{valid}(x)}$ (e.g., $L_2$-boundedness or $k$-hot) and retain: (i) the encoding-validity proof $\pi_{\text{Enc}}(sk, x)$, which in PRG-SecAgg certifies the linear relation $y = sk + Gx$; (ii) the range/smallness proof $\pi_{0 \le x < t}$, which is necessary to rule out plaintext wrap-around and to enable decoding via $G^{-1}$ after aggregation; and (iii) the distributed key-correctnessprotocol that proves $\sum_i sk_i = sk$ without extra interaction rounds. This matches the "minimal but deliverable" ACORN configuration required for correctness and decodability, while excising input-validity checks that are not needed by our threat model.

**Quantization, packing, and message schedule.** Unless otherwise noted, we use $t = 2^{16}$ to mirror typical FL-style 16-bit quantization. The input length $\ell$ equals the task/model dimension. Each client commits to $x$ and $sk$ via vector Pedersen commitments and sends $(y, \text{com}_{sk}, \text{com}_x, K, \pi_{\text{Enc}}, \pi_{0 \le x < t})$. The server batch-verifies proofs before mask/key recovery.

## N ALGORITHM

As shown in Figure 7, the process framework of SVAFD is presented. The first round is called "Knowledge Aggregation" and it involves interaction between clients shown in Algorithm1. The second round Knowledge Verification involves interaction between clients and servers, which in shown in Algorithm2.

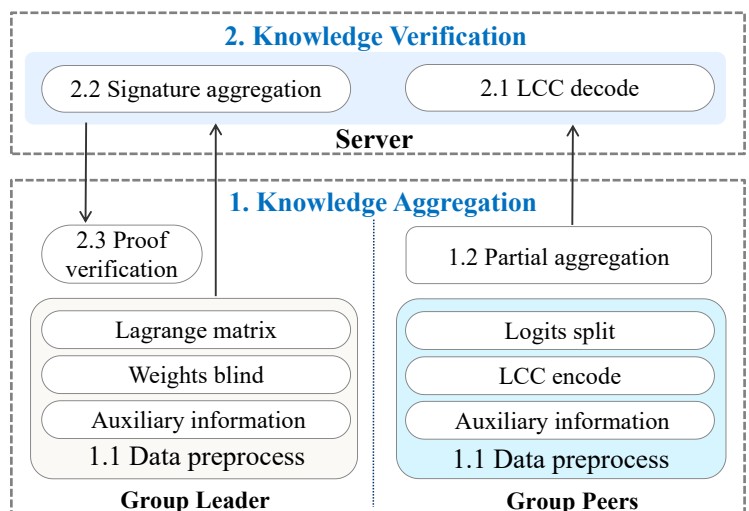

Figure 7: Workflow of SVAFD.

## O USE OF LLM

LLM Usage Disclosure. We used the large language model (GPT-5) solely for language editing of this manuscript—namely grammar, phrasing, and clarity. The tool did not generate ideas, design experiments, analyze data, write code, or draft substantive content; all scientific contributions are the authors' own. All edits were manually reviewed and approved by the authors, who take full responsibility for the final text. Only non-sensitive draft manuscript text was provided to the tool; no confidential, personal, or proprietary data were shared. This usage complies with the conference's policies on responsible use of generative AI.

