# OpenReview forum: "It takes two for security: A Verifiable Co-Aggregation Protocol for Heterogeneous Federated Distillation"
_ICLR.cc/2026/Conference — ICLR 2026 Conference Withdrawn Submission_

### Official Review · Reviewer_x5vo · 2025-10-31

**Soundness:** 4
**Presentation:** 3
**Contribution:** 4
**Rating:** 6
**Confidence:** 4

**Summary:**

Support for arbitrary polynomial aggregation $\mathcal{h}(\cdot)$ of degree $\deg(\mathcal{h})$.

Key technical contributions:
--MM-LCC (Multi-to-Multi Lagrange Coded Computing) enabling homomorphic share aggregation.
--Two-stage co-aggregation with client-side privacy-critical ops and server-side coordination.
--Verifiable proofs with low client overhead.
--Empirical results show 1.32× lower computation than Armadillo (NeurIPS’24) and 2.37× less client-side workload, with robustness to 10% dropout and 10% malicious clients.

**Strengths:**

Originality: Co-aggregation paradigm + MM-LCC is a creative and principled synthesis of coded computing and secure aggregation.
Quality: Sound theoretical foundation (Theorems, Lemmas) with matching experiments. Outperforms SOTA on efficiency, client load, and robustness.
Clarity: Core idea (client-side privacy, server-side coordination) is intuitive and well-motivated via Figure 1.
Significance: Reduces client compute by >2× vs. Armadillo; enables secure FD at scale. Broadly applicable beyond FD (e.g., secure gradient aggregation).

**Weaknesses:**

Missing Appendices: Privacy proof (Appendix K), verifiability proof (Appendix J), and full dataflow (Figure C) are not provided. This limits full assessment of security claims.
Complex-domain LCC error analysis is brief. Real-world floating-point precision may amplify errors in decoding.
Verification cost breakdown is missing. Claimed “<5ms” but no per-client timing.
Peer set construction is random — orthogonal, but no discussion of impact (e.g., clustering-based peers).

**Questions:**

The privacy proof (Appendix K) claims T-Privacy via truncated Gaussian blinding. Can you provide the reduction to DLP or CDH and the distinguishability bound?
How does decoding error scale with $K$, $T$, and floating-point precision? Is there a regime where $\hat{Y} \neq Y^*$ within machine precision?
The verifiability requires the leader to receive all $\Upsilon_i$. How is this done securely? Is there a key exchange phase?
Can SVAFD support asynchronous aggregation (straggler mitigation beyond dropout)?
Any overhead comparison with SecAgg++ or TFLite-based FD in real mobile deployments?

---

### Official Review · Reviewer_7vja · 2025-10-31

**Soundness:** 3
**Presentation:** 2
**Contribution:** 3
**Rating:** 4
**Confidence:** 2

**Summary:**

This paper addresses the issue of low efficiency in privacy-preserving aggregation protocols in federated distillation by proposing a federated collaborative aggregation protocol named SVAFD. This method divides the aggregation process into local partial aggregation encoding and server-side global decoding, aiming to balance the responsibility boundaries between clients and servers, thereby effectively protecting the privacy of shared logits in federated distillation. This multilateral client/server co-aggregation approach enhances the security and reliability of the federated system. Experimental results show that this method significantly reduces the consumption of computational resources.

**Strengths:**

The multilateral client/server co-aggregation design of this method demonstrates innovation and effectively ensures the security and privacy protection of federated distillation data. The article provides a thorough theoretical analysis of the method, with rigorous logic and clear structure. The author elaborates on the fundamental principles of each component, making it easier to understand the effectiveness of the method in depth.

**Weaknesses:**

The comparative experimental results are somewhat lacking. Adding comparisons with other methods addressing federated communication security, or including additional experimental results for higher security guarantees, would help enhance the paper's persuasiveness. The expression of the method in the article is relatively difficult for readers to understand. The numerous formulas and symbols in the text, along with extensive cross-references to appendices, make this article difficult to understand. It is recommended that the logical expression of the article be revised.

**Questions:**

1. Where is the specific implementation of the local partial aggregation design? Equation 4 seems to still compute all logits from the clients. On the other hand, how are the clients selected for aggregation? Which clients need to perform partial aggregation? What does the distributed collaboration process look like among the clients? What does $\overline{Y} $ represent on the server? The server obtains aggregated shared information; how is this used to decode the data from partial aggregation? The $\widehat{Y} $ in line 288 and Figure C(d) need to be revised.

---

### Official Review · Reviewer_SMWq · 2025-10-31

**Soundness:** 3
**Presentation:** 3
**Contribution:** 3
**Rating:** 4
**Confidence:** 5

**Summary:**

This paper proposes SVAFD, a secure and verifiable protocol for heterogeneous Federated Distillation (FD). It solves the "Aggregation Subject Dilemma," where existing methods are either vulnerable (server-dominated) or inefficient (fully distributed). SVAFD introduces a "co-aggregation" model: clients perform lightweight, privacy-critical local aggregation, while the server handles heavy computation like global decoding and generating aggregation proofs. This hybrid approach achieves a balance, providing strong privacy and verifiability while remaining efficient and resilient to client dropouts.

**Strengths:**

1.	It breaks the dichotomy of traditional centralized (server-dominated) and decentralized (server-free) approaches by redefining the responsibility boundaries between the server and clients, achieving a principled balance between privacy and efficiency.
2.	SVAFD ensures client privacy even against a server colluding with a subset of clients , offers an unforgeable aggregation proof for process verifiability , and is resilient to client dropouts (stragglers).
3.	SVAFD is highly efficient, completing aggregation in just two rounds. It strategically offloads computationally intensive tasks, such as global decoding and signature aggregation, to the resource-rich server.

**Weaknesses:**

1.	The paper's threat model extensively covers malicious servers or peers, but it's less clear what happens if the Leader itself is malicious or fails, which could introduce a new single point of failure.
2.	The experiments are conducted with a maximum of 200 clients. While this validates the protocol's mechanics, it doesn't confirm its scalability for large-scale FL scenarios (e.g., thousands of clients).
3.	SVAFD is a complex system that combines several advanced techniques, including an extension of Lagrange Coded Computing (MM-LCC) and cryptographic mechanisms like bilinear pairings. This high degree of complexity would likely make real-world deployment.

**Questions:**

I will reconsider my score during the rebuttal phase based on the authors' response to following issues.

1.	What are the differences between secure aggregation in Federated Learning and that in the proposed Federated Distillation scheme? Can the secure aggregation methods used in Federated Learning be directly applied to Federated Distillation?
2.	The paper mentions two types of aggregation schemes, namely server-free and server-dominated architectures. However, to my knowledge, there also exist dual-server-based aggregation schemes that support verifiable aggregation results (e.g., (1) ELSA: Secure Aggregation for Federated Learning with Malicious Actors. (2) Janus: Dual-Server Multi-Round Secure Aggregation with Verifiability for Federated Learning). How does the proposed scheme compare with such approaches in terms of advantages? A citation and comparison with this type of scheme should be added to the paper to provide a more comprehensive discussion.
3.	This paper involves a substantial amount of cryptographic notation. It is recommended to include a notation table in the appendix to improve readability.
4.	Figure 2 presents the system workflow. It is recommended to add numbering to indicate the sequential order of the steps.
5.	In line 220, $\sum_{i} w_i = 1$. The generation of $w_i$ appears too simplistic. If the weights are not properly assigned, it may directly affect the training performance. Typically, these weights are correlated with the size of each user’s local dataset.
6.	In line 255, the phrase “sufficient numbers of” is ambiguous. It is unclear what “sufficient” refers to—does it imply reaching a specific quantitative threshold, or is it based on a subjective judgment? Such wording is not formal and should be replaced with a more precise description or explicit criterion.

---

### Note · Authors · 2025-12-03

I have read and agree with the venue's withdrawal policy on behalf of myself and my co-authors.